# Development and Evaluation of a Massive Open Online Course on Healthcare Redesign: A Novel Method for Engaging Healthcare Workers in Quality Improvement

**Mitchell Dwyer [1],\*, Sarah J. Prior [2], Pieter Jan Van Dam [3], Lauri O'Brien [1] and Phoebe Griffin [4]**

1   Tasmanian School of Medicine, University of Tasmania, Hobart, TAS 7000, Australia
2   Tasmanian School of Medicine, Rural Clinical School, University of Tasmania, Burnie, TSA 7320, Australia
3   School of Nursing, Cradle Coast Campus, University of Tasmania, Burnie, TSA 7320, Australia
4   Tasmanian School of Medicine, University of Tasmania, Launceston, TSA 7000, Australia
\*   Correspondence: mitchell.dwyer@utas.edu.au

**Abstract:** Abstract: IntroductionHealthcare workers are under increasing pressure to use limited resources more efficiently and improve patient outcomes. Healthcare redesign, a quality improvement methodology derived from the automotive industry, is a proven means of achieving these goals. Continuing Professional Development (CPD) opportunities for nurses seeking to build their capacity for healthcare redesign often come in the form of university courses, which can be costly and prohibitively time-consuming. We developed a Massive Open Online Course (MOOC) with a view to increasing the number of healthcare workers undertaking CPD in healthcare redesign and subsequently using these principles in their workplaces. The aim of the current study is to describe the development of our MOOC and its initial feedback from users. **Materials and Methods:** The theoretical and practical components of an existing postgraduate award course unit were made fit for purpose by being arranged into six weekly modules, before being transposed to an established learning management platform for MOOCs. Related quizzes, videos and interactive activities were then developed and included in each of these modules. Peer review of this content was completed by subject matter and teaching and learning experts prior to the MOOC being launched. **Results:** After running for nine months, 578 participants had enrolled in the MOOC, of whom 118 (20%) had followed through to completion. Participants were overwhelmingly from Australia (89%) and identified as female (78%). Preliminary feedback obtained from participants was positive, with 81% of respondents agreeing that they were satisfied with their experience, and 82% intending to apply their knowledge in practice. **Conclusions:** The MOOC has addressed a learning need by providing a brief and free form of education; learning from its development will help others seeking similar educational solutions. Initial feedback suggests the MOOC has been well-received and is likely to be translated into practice.

**Keywords:** massive open online course; healthcare redesign; quality improvement; lean; six sigma; clinical engagement; continuing professional development

## 1. Introduction

Healthcare workers are under increasing pressure to make more efficient use of limited resources and to improve patient and organisational outcomes [1,2]. Healthcare redesign, a quality improvement methodology, is a means of achieving these goals through reducing waste and unwarranted variation [3]. "Waste" in a healthcare context can occur in a number of forms and essentially relates to any activities which are not value-adding; this may include things such as ordering unnecessary diagnostic tests or stockpiling supplies which are seldom used [4]. When waste in any form is reduced, resources can be diverted to more needful areas to improve patient care (e.g., a substantial cost saving could allow a hospital to increase the number of surgical procedures it performs). Unwarranted variation refers to

"wide variations that cannot be explained by illness severity or patient preference" [5]. For example, if, with individual patients' characteristics considered, hospital A is far less likely than hospital B to provide a gold standard treatment for a given condition, this would be evidence of unwarranted variation. There are numerous reasons why unwarranted variation is problematic to the safety and quality of patient care, which centre around equity of access to services and the potential for harm resulting from unnecessary tests or treatments [6].

Learning opportunities for nurses seeking to build their redesign capacity and capability are often in the form of paid courses, which require significant financial and time investments [7–9]. Barriers to advancing knowledge in the healthcare redesign field include these economic factors, as well as a lack of support or sponsorship from health systems and services [10]. However, there is evidence to suggest that healthcare organisations are now beginning to promote and support opportunities for learning around quality improvement and health service improvement for staff [11]. Therefore, there is a need to explore novel healthcare redesign educational opportunities for busy nurses and other healthcare workers, who may lack the time, funding and current organisational support to pursue existing award course offerings.

In Australia, the Australian Health Practitioner Regulation Authority (AHPRA), a national agency, focusses on improving the standards and quality of health services [12]. The AHPRA supports 15 national practitioner boards that regulate the health professions as set by the Health Practitioner Regulation National Law (the National Law) [13]. Under the National Law, all registered health practitioners must undertake Continuing Professional Development (CPD). CPD relates to how health practitioners maintain, enhance and broaden their knowledge and expertise to provide appropriate and safe care (AHPRA, 2019). One of the Boards, the Nursing and Midwifery Board of Australia, stipulates that registered nurses and midwives need to undertake 20 h of CPD per year to meet their registration standard [14].

Massive Open Online Courses (MOOCs), which offer free, time-unlimited access to virtual learning environments, may provide a means of addressing barriers to healthcare redesign education, as well as an opportunity for nurses to undertake CPD. According to Class Central, the leading MOOC aggregator site, there are now in excess of 19,000 MOOCs being offered by 950 universities, reaching some 220 million participants worldwide, representing a huge growth since record-keeping began approximately ten years ago [15]. This period has also seen several criticisms levelled against MOOCs as a platform. Indeed, low completion rates, the need for digital literacy (and the potential for this to exclude some learners) and the tendency to use outdated, didactic teaching methods (albeit using the latest technology) are often-cited concerns [16–18].

The proliferation of MOOCs has also given rise to an equally large number of MOOC formats and topics, making it difficult to study the longer-term efficacy and suitability of MOOCs as a whole. It therefore makes sense that MOOCs should be evaluated independently to determine their value to participants. We developed a MOOC in the niche area of healthcare redesign with a view to increasing the number of healthcare workers undertaking CPD in this field and subsequently using redesign principles in their workplaces to improve health systems and services and, ultimately, patient outcomes. The current study seeks to describe the development of this MOOC and to evaluate the MOOC from the perspective of the initial cohort of participants who completed it, for the purpose of quality improvement.

## 2. Materials and Methods

### 2.1. Designing the Course and Methods

The impetus to create this MOOC came from the realisation that many nurses who would like to complete the University of Tasmania's (UTAS) Graduate Certificate in Clinical Redesign [7] lacked the time, resources, and organisational support to do so. The Graduate Certificate is a postgraduate award course comprising four units, two of which are project-

based and require students to undertake a redesign project within their own workplaces. The remaining two units are in a traditional didactic format, providing students with a foundation in redesign principles and translational research. The four units of the Graduate Certificate are typically completed over the course of four consecutive semesters or 1.5 years. The MOOC was not created to replace the existing award course, but rather to complement it by providing an accessible introduction to the award course's subject matter. The Tasmanian Health Service (THS), the local acute health service and a major industry partner, also expressed a wish for an introductory-type course that required less of a time commitment from students than a postgraduate award course.

Our "Healthcare Redesign MOOC" was therefore developed as a free, six-week course designed to guide participants through the principles of health service improvement utilising healthcare redesign methodology. Subject matter experts and lecturers in health services improvement collaborated to develop six modules which were derived from the aforementioned award course [7]. A summary of these modules and their Intended Learning Outcomes (ILOs) can be found in Table 1. The six modules were then transposed to a pre-existing Learning Management System (LMS) developed by the Wicking Dementia Research and Education Centre, a partner organisation which has itself developed several highly successful MOOCs [19,20].

**Table 1.** Summary of Modules in the Healthcare Redesign MOOC.

| Modules | Intended Learning Outcomes |
|---|---|
| Welcome | Course introduction, glossary of terms, pathways to further study |
| Module One: The Case for Change<br><br>Module Two: Theories for redesigning Healthcare | 1. Demonstrate an understanding of how healthcare redesign can improve patient/consumer experience, health service performance and health outcomes.<br>2. Identify key concepts and approaches used in healthcare redesign to improve health services. |
| Module Three: Techniques for Engaging People | 1. Describe frameworks and models to the relational challenges inherent in the redesign process. These challenges include, motivating staff in a change-fatigued environment, engaging busy clinicians, generating shared solutions, dealing with negativity and facilitating groups effectively.<br>2. Enhance communication and engagement skills related to work culture and redesign.<br>3. Enhance skills in motivating teams and facilitating change. |
| Module Four: Understanding the Problem | 1. Identify key concepts and approaches used in healthcare redesign to improve health services.<br>2. Apply redesign concepts and approaches to identify process problems in a healthcare setting and construct a plan to address these. |
| Module Five: Addressing the Problem | 1. Identify key concepts and approaches used in healthcare redesign to improve health services.<br>2. Compare ways of engaging people in healthcare redesign to improve health services. |
| Module Six: Evaluation and Sustainability | 1. Demonstrate an understanding of how healthcare redesign can improve patient/consumer experience, health service performance and health outcomes.<br>2. Identify key concepts and approaches used in healthcare redesign to improve health services. |

Note: a quiz score of at least 70% was needed to pass one module and advance to the next.

Each of the six modules employs a variety of teaching methods, such as didactic text, videos and case studies, along with links to relevant articles for further reading. Each module also includes a host of formative learning activities, such as interactive quizzes and reflective exercises. At the conclusion of each module, there is a formal, summative quiz which must be passed (>70%) to progress to the next module. These quizzes relate specifically to the preceding module's content and consist of 4–5 questions in a true/false or multiple-choice format, with no limit on the number of attempts participants can make. Each module equates to approximately 3 h of work, and once all are completed, deemed equivalent to 18 CPD hours, they represent a large portion of the 20 CPD hour per annum requirement for Australian registered nurses [14]. Participants can complete these units at their own pace, with no time restrictions on the availability of the modules. Upon completing the MOOC, participants have the option to purchase a certificate of completion in either digital or hardcopy plus digital format.

The target audience of the MOOC is similar to that of our postgraduate award course [7], namely nursing staff, allied health practitioners, medical officers and administrative staff. The MOOC was designed to cater for the local Tasmanian audience but is still relevant to national and international participants. There is, however, no pre-requisite knowledge for enrolling in the MOOC, and it is suitable for anyone motivated by positive change and quality improvement within healthcare. Prior to its launch, the MOOC underwent a peer review process within the authors' university and by external subject matter experts from the authors' wider professional network. With several amendments made following this feedback, the MOOC was launched on 1 July 2021. Promotional material was then distributed to several Australian healthcare providers, and the MOOC continues to be advertised by UTAS. To further increase its visibility, "share" buttons were created, which allow participants who complete the MOOC to easily share news of their achievement via their Facebook and LinkedIn profiles.

### 2.2. MOOC Evaluation

### 2.2.1. User Analytics

General demographic information was captured through the enrolment process and exported from the LMS, namely the participants' country of residence, age and gender. Participants were also able to inform as to whether they were employed by a select group of UTAS partner healthcare organisations; however, these data did not begin to be collected until February 2022. Data relating to time spent on individual modules, course completion and the purchase of certificates were also extracted from the LMS.

### 2.2.2. Participant Experience Survey

Participants who completed the MOOC were invited to anonymously complete a brief survey about their experiences of the course. A link from the LMS directed participants to the survey, which was delivered via a LimeSurvey platform [21]. Following a brief statement about the rationale for the survey, consent was assumed by the completion of the survey. The survey comprised six quantitative Likert-type questions and eleven free-text questions, which asked participants to rate and comment on various aspects of their experience, including their learning and their overall experience of the course.

### 2.2.3. Data Analysis

Microsoft Excel was used to organise and analyse the data. Participants' baseline characteristics were compared between those who had completed the MOOC and those who had not, using $\chi^2$ tests within IBM SPSS. Where deemed appropriate, Fisher's exact tests were also conducted using the web-based open-access tool Astatsa Online Web Statistical Calculators (Navendu Vasavada, astatsa.com (accessed on 10 June 2022)). Given the relatively small population of individuals who have completed our MOOC (N = 118), a response rate of N = 83 (70%) to our Participant Experience Survey would allow us to have 90% confidence that this sample was representative of this population, assuming

a 5% margin of error (Sample Size Calculator, available at Calculator.net (accessed on 10 June 2022)). Qualitative data were analysed using conceptual content analysis, supported by quotes selected to illustrate the breadth of participants' experiences.

## 3. Results

### 3.1. Participant Characteristics

As of mid-April 2022, the Healthcare Redesign MOOC had 578 enrolments, of whom 118 (20.4%) completed the course. The majority of participants were female (78%) and most were in the age groups of 30–39 years (29%) and 40–49 years (29%). Most participants were from Australia (88%) and 33% were employed within the THS (Table 2). The characteristics of participants who had completed the MOOC did not differ markedly from those who were yet to complete the MOOC.

**Table 2.** Participant Characteristics.

| Variable | | Participants yet to Complete the MOOC N = 460 | | Participants who Completed the MOOC N = 118 | | *p*-Value |
|---|---|---|---|---|---|---|
| | | Frequency | Percent | Frequency | Percent | |
| Gender | | | | | | |
| | Female | 375 | 81.5% | 83 | 70.0% | |
| | Male | 85 | 18.0% | 35 | 30.0% | 0.010 |
| Age Group † | | | | | | |
| | 0–19 years | 7 | 1.5% | 1 | 0.8% | |
| | 20–29 years | 72 | 15.7% | 23 | 19.5% | |
| | 30–39 years | 142 | 30.9% | 29 | 24.6% | |
| | 40–49 years | 125 | 27.2% | 36 | 30.5% | |
| | 50–59 years | 85 | 18.5% | 20 | 16.9% | |
| | 60–69 years | 26 | 5.7% | 4 | 3.4% | |
| | 70–79 years | 1 | 0.2% | 2 | 1.7% | 0.286 |
| Country | | | | | | |
| | Australia | 401 | 87.2% | 105 | 89.0% | |
| | New Zealand | 11 | 2.4% | 2 | 1.69% | |
| | India | 5 | 1.1% | 1 | 0.85% | |
| | Great Britain | 3 | 0.7% | 2 | 1.69% | |
| | Other | 40 | 8.7% | 8 | 6.78% | 0.738 |
| Employer * †† | | (N = 148) | | (N = 37) | | |
| | Tasmanian Health Service | 47 | 31.8% | 12 | 32.4% | |
| | No answer | 30 | 20.3% | 6 | 16.2% | |
| | Other | 28 | 18.9% | 8 | 21.6% | |
| | Calvary Care | 11 | 7.4% | - | - | |
| | Withheld | 32 | 21.6% | 11 | 29.7% | 0.421 |
| Certificate Purchases | | | | | | |
| | None | - | - | 85 | 72.0% | |
| | Digital | - | - | 31 | 26.3% | |
| | Printed | - | - | 2 | 1.7% | - |

† N = 5 participants did not provide their date of birth. * Data were only collected from 1 February 2022 onwards. †† The names of employers with <10 participants were withheld to protect the privacy of these participants.

### 3.2. Participants' Attempts at Completing Module Quizzes

The number of individual participants who attempted the module quizzes decreased with each successive module, with over half of those enrolled yet to attempt the first module's quiz (Table 3). Each quiz was re-attempted by participants who had already passed the quiz and hence had no need to redo it in order progress to the following module. This was most evident in the Module 1 quiz, where 220 participants had 409 successful attempts at the quiz.

**Table 3.** Participants' Attempts at Completing Module Quizzes (N = 578).

| Quiz | Participants Attempting | Attempts | Successful Attempts (n, %) | Mean No. of Attempts Per Participant |
|---|---|---|---|---|
| Module 1 | 220 | 601 | 409 (68.1) | 2.73 |
| Module 2 | 179 | 229 | 203 (88.6) | 1.28 |
| Module 3 | 161 | 249 | 221 (88.8) | 1.55 |
| Module 4 | 148 | 168 | 163 (97) | 1.14 |
| Module 5 | 139 | 247 | 199 (80.6) | 1.78 |
| Module 6 | 118 | 167 | 147 (88) | 1.42 |

### 3.3. Participant Experience Survey

Results from the Participant Experience Survey are divided into three sections: (1) Satisfaction with the MOOC, (2) Motivation for Enrolling in the MOOC and (3) Translation of Learning. Each of these sections is described in more detail below.

### 3.4. Participants' Satisfaction with the MOOC

Most participants (81%) either agreed or strongly agreed that they were satisfied with their MOOC learning experience, with the same percentage also stating that they would recommend the MOOC to others. An improved understanding of healthcare redesign methodology was reported by 81% of participants, with 82% planning to apply their new knowledge. Sixty-two percent of participants indicated that they would like to undertake further online university study based on their experience with the MOOC (Table 4). Participants provided a range of reasons for their satisfaction with the MOOC with most centred around the way in which the information was delivered and the course outcomes in terms of new knowledge and improved understanding.

**Table 4.** Participant Experience Survey Questions including Perception of the Quality of the MOOC.

| Question | Strongly Disagree | Disagree | Neutral | Agree | Strongly Agree | No Answer | Total Respondents (N = 118) |
|---|---|---|---|---|---|---|---|
| I was satisfied with my MOOC learning experience | 3 (4%) | 0 | 1 (1%) | 35 (42%) | 32 (39%) | 12 (15%) | 84 (71%) |
| I would recommend the MOOC to others | 1 (1%) | 0 | 1 (1%) | 27 (33%) | 40 (48%) | 14 (17%) | 83 (70%) |
| My understanding of healthcare redesign methodology has improved | 1 (1%) | 0 | 1 (1%) | 34 (41%) | 33 (40%) | 14 (17%) | 83 (70%) |
| I plan to apply the knowledge I have gained from the MOOC | 1 (1%) | 0 | 0 | 33 (40%) | 35 (42%) | 14 (17%) | 83 (70%) |
| I would like to do further online university study | 1 (1%) | 1 (1%) | 16 (19%) | 27 (33%) | 24 (29%) | 14 (17%) | 83 (70%) |
| | Poor or very poor | Average | Good | Very good | Excellent | No answer | Total Respondents (N = 118) |
| How would you rate the quality of this MOOC overall? | 0 (0%) | 0 (0%) | 13 (16%) | 22 (27%) | 32 (39%) | 16 (19%) | 83 (70%) |

*"I was very Satisfied with the MOOC because it was very well explained and descriptive by the tutors and professionals in the area of Healthcare Redesign. It was an excellent course!!!!! The videos were excellent in teaching the subject".*P13

*"I found the balance of written and video presentation of information very user friendly. I appreciated the flexibility in how I timed the completion of the course".* P18

*"It has given me a better understanding of the process of which the health system can continually improve and work together as a team to resolve issues in the workplace but also improve in different areas but we all are on the same page to have a harmonious workflow by working together no matter which area we work in".* P53

*"It was a very informative MOOC however is it not as engaging with the audience. Including for user input activities would in my opinion make it better".* P78

Free text comments also highlighted that participants thought the best things about the MOOC were the online study design (48%), the content (42%), the way in which the MOOC was delivered (33%) and the free access (15%).

*"How easy the course is to navigate through due to the layout and course information provided".* P82

*"Quick, free access to relevant knowledge to help make change in TAS Health that benefits patients and makes better use of existing resources".* P20

Participants generally did not distinguish between modules in terms of their relative value, with 27/40 (68%) suggesting that they enjoyed the course as a whole. However, some participants highlighted that a particular module was more beneficial for them in relation to their current role or their interest in healthcare redesign.

*"Stakeholder engagement—the previous work I did didn't really involve that, so I found it really helpful having tips for successful engagement".* P24

*"Module 2: Theories for Re-designing Healthcare. I especially liked the Lean Thinking 5 Core Principles and the focus on reducing waste and improving flow".* P74

*"Sustainability. I have already been involved in clinical redesign processes however they haven't all been sustainable".* P66

Participants were provided an opportunity to describe, in free text, any further activities or discussion questions that they felt would have improved the content or other approaches for delivery that may have enhanced their experience in the MOOC. There were a number of comments around the examples included in the MOOC that were based on the acute healthcare setting rather than the primary health setting.

*"Would be nice to have more situations as an example. At least two or three of different settings".* P15

*"It was appropriate it would be great to compliment this Program with an Excellence in Pathways Program through the RHH Centre for Education and Research Nursing and Midwifery. This would be beneficial for every manager in Health care. The length and timeframe perfect, I would have loved a booklet of the pages of the program that I could use as a reference nothing major just a little prompt—I am planning of making my own and using it as a prompt".* P28

*"I would have like one or two more examples of a completed plan from developing the idea to implementing and assessing the outcomes".* P65

Additional free text comments from 19 participants around the video course content included comments related to the video clips being beneficial (12/19) and that the content of the video clips could have been improved (overseas footage (1), too long (1), links not working (1) and only one healthcare setting (1)).

*"Video clips assisted me greatly in having a clear view and picture of how clinicians deal with everyday life at a Clinic in helping the work to better flow and focusing on do their best to help the patients and provide the best service for them".* P13

*"Examples of different health settings, e.g., a lot of examples were hospital based".* P31

*3.5. Motivation for Enrolling in the MOOC*

Fifty-three (45%) participants responded when asked to describe the reason for choosing and enrolling in the Healthcare Redesign MOOC. The three main categories evident

in the free text answers included knowledge attainment (36%), personal or professional interest (32%) and career progression (19%). Specifically, participants were interested in improving outcomes for their patients or the quality of their own health services, undertaking free, relevant education, developing their change management skills and refreshing their current knowledge.

> *"Currently have a large turnover of staff in the department I work, this is very disappointing considering the stable workforce, we had 5–10 years ago. It has highlighted to me that patient care has changed but work processes currently don't align with these significant changes".* P74

> *"Want to effect major change in addressing adaptive health issues in primary healthcare, ambulance service and hospital services. I thought this might help understand how to effect change and get buy in".* P20

> *"The fact it was free but also that I do notice things that could be improved in the hospital and do bring them up but I was interested in learning how to approach implementing improvement initiatives".* P36

> *"Personal interest I would like to do this work".* P56

> *"Wanting to understand clinical redesign. Also working within XXX frustrated with inefficiencies and lack of change".* P85

*3.6. Translation of Learning*

Seventeen (15%) participants provided free text answers concerning the application of their MOOC learnings. Three participants suggested that they have utilised their new knowledge, mostly through the reinforcement of prior knowledge and planning for a quality improvement/redesign project. Other participants indicated that they had not yet had a chance to apply their new knowledge or that it was not applicable within their current role.

## 4. Discussion

Health service improvement or healthcare redesign is a high-pressure, rapidly evolving area, which is of great relevance to industry and government for economic, equity and quality of care reasons [22]. Although a large number of short industry training courses are available in the area of health service improvement and system innovation, there is little in the way of free courses specifically focused on healthcare redesign [23]. The Healthcare Redesign MOOC filled the important purpose of organisational learning, as organisations are starting to focus on supporting learning among employees, promoting innovation, reducing waste and improving efficiency [11]. This, coupled with emerging evidence that more organisations are using MOOCs to develop employees' skills to carry out their work [24], suggests our educational strategy is on target. As educators, we took industry needs into consideration, as the Healthcare Redesign MOOC was designed to aid the development of sustainable Tasmanian and national healthcare systems, focussing on ongoing improvement in the quality, effectiveness and safety of care delivery and inspiring widespread engagement with the process. This strategy was further supported by MOOCs in the workplace having a positive impact on job competency and innovation [25].

Many participants identified themselves as THS employees, and this might have been a result of discussions with THS leaders, who were committed to recommending the MOOC to their staff members. This finding is in line with other organisations starting to support their employees' development by recommending they enrol in MOOCs [26]. There is not only support from organisations for MOOCs; many educators are also starting to embrace the concept of MOOCs, arguing that these learning platforms contribute to significant benefits to both learners and educators [27]. Increasing the knowledge of participants and building the reputations of educators as experts in the field are some of the main benefits reported [28]. Increasing the reputation of educators was reflected in some of

the qualitative data, whereby the educators were regarded as professionals in the area of healthcare redesign.

The high completion rate of 20% compared to the typical rates of completion of between 5–10% [18,29] is a strong indicator that the course was well-designed and relevant to its participants. The course was created with quizzes at the end of each module, which helped prevent an ineffective learning process through the detection of missed activities [30]. Moreover, participants did not receive a certificate of completion if they had not passed all assessment tasks. The high quality of the MOOC is supported by our findings that 81% of the participants who completed the MOOC were satisfied with the MOOC experience and items, such as online study design, the content and the way in which the MOOC was delivered, which were highly rated. Additionally, two-thirds of respondents described the overall quality of the MOOC as being "very good" or "excellent" with a further 16% describing it as "good" (Table 4). However, it became clear that more work could be undertaken to make the MOOC more engaging and to create user input activities. This was evident in the large number of participants who had not yet attempted the first module's quiz and the decrease in quiz attempts for each successive module. This decline in engagement is likely due in part to an absence of interaction with peers and the course facilitators, as has been suggested by several authors [31–33]. Successful MOOCs use a wide variety of interactive learner activities and implementing these activities would strengthen the MOOC design [34,35].

There were a number of comments around the examples included in the MOOC being based on the acute healthcare setting only. Until recently, there was only a limited number of academic works available discussing healthcare redesign in a primary health setting; however, evidence is emerging that healthcare redesign can be a successful methodology in primary health [36,37] and therefore these lessons should be shared. It was pleasing to find that many participants were interested in improving outcomes for their patients or the quality of their own health services. This, in a way, was not a surprising finding, as many healthcare professionals are motivated to choose their profession by a desire to help others, and it has been argued that this desire to help others is one of the most important factors that influenced career choices among students in nursing, medicine and pharmacy [38].

Most of the participants' responses were positive and reflected learning, as the improved understanding of healthcare redesign methodology was reported, with many planning to apply their new knowledge. This is supported by the fact that many participants chose to re-attempt quizzes they had already passed (with no requirement to do so), presumably to refine or reinforce their knowledge. However, it is challenging to make a direct link between the MOOC and potential successful projects participants might choose to undertake [8]. It has been previously argued that to grasp the skills and knowledge of healthcare redesign, a pedagogy that combines didactic theoretical learning with experiential, project-based learning and coaching needs to be part of educational offerings in the area of healthcare redesign [39]. One of the aims of the MOOC was to inspire a widespread engagement with the process of healthcare redesign with an encouragement to undertake further practical studies in the area. Based on the positive findings of this study we anticipate that many participants will consider further study in healthcare redesign.

Our study contains some limitations which must be acknowledged. Firstly, our Participant Experience Survey was only completed by participants who had completed the entire MOOC. Hence, we were not able to gain the perspectives of individuals who had commenced the MOOC and had since become disengaged. Future studies of this kind are encouraged to seek out the perspectives of such participants (e.g., via some form of exit survey), as it may yield insights on how to maintain participants' engagement in the MOOC. Secondly, whilst the authors included alternate text for all images, it is possible that some participants experienced difficulty accessing and navigating the MOOC's content, which could have been mitigated by conducting usability testing of the platform prior to the MOOC's release. Thirdly, given that our target audience is likely to have had some exposure to redesign concepts prior to undertaking the MOOC, the use of a pre-post

measure of participants' knowledge, as suggested by Alturkistani et al. [40], would have added rigour to the evaluation process. Lastly, our participants were also predominantly female, middle-aged and Australian, which may impact the generalisability of our findings to other settings.

**Author Contributions:** Conceptualization, M.D. and P.J.V.D.; methodology, S.J.P.; writing—original draft preparation, P.G.; writing—review and editing, M.D., L.O., S.J.P. and P.G. All authors have read and agreed to the published version of the manuscript.

**Funding:** This research received no external funding.

**Institutional Review Board Statement:** The study was approved by the University of Tasmania Human Research Ethics Committee (Project ID: 27239).

**Informed Consent Statement:** Participant consent was waived due to the anonymity of participants' responses and the negligible risk of potential harm to participants.

**Data Availability Statement:** Microsoft Excel was used to organise and analyse the data. Participants' baseline characteristics were compared between those who had completed the MOOC and those who had not using $\chi^2$ tests within IBM SPSS. Where deemed appropriate, Fisher's exact tests were also conducted using the web-based open-access tool Astatsa Online Web Statistical Calculators (Navendu Vasavada, astatsa.com (accessed on 10 June 2022)). Qualitative data were analysed using conceptual content analysis, supported by quotes selected to illustrate the breadth of participants' experiences.

**Acknowledgments:** Suzanne Waddingham, Tasmanian School of Medicine, UTAS; Joshua Eastgate, Wicking Dementia Research and Education Centre, UTAS; Chris Parker, Wicking Dementia Research and Education Centre, UTAS.

**Conflicts of Interest:** The authors declare no conflict of interest.

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
