# Peer review of "Development and Evaluation of a Massive Open Online Course on Healthcare Redesign: A Novel Method for Engaging Healthcare Workers in Quality Improvement"

_nursrep, doi:10.3390/nursrep12040082_

Round 1
Reviewer 1 Report
This manuscript presents very interesting data on Massive Open Online Course on Healthcare Redesign in Australia that is clearly of broad interest and deserves publication. This is a "delicious" paper I have no hesitation in recommending it for publication following some minor tidy-up.
Page 9 line 15
The high completion rate of 20% compared to typical rates of completion of between 5-10% (18, 29) is a strong indicator that the course was well-designed and relevant to its participants.
It would be useful to include a discussion about the information that only 20% of the participants completed the entire course.
It would be useful to include this information the following pint; as e-learning methods, for example, synchronous e-learning training, lectures are currently being developed in which participants and instructors discuss interactively. I think it is appropriate to mention this point. .
Author Response
Thank you for your edit of our manuscript. Please find attached our response to your comments.

Reviewer 2 Report
Dear authors
I find this paper very interesting and above all pertinent.
I have several questions or comments
Major issues
The objective of the paper is very clear but I am not clear about the hypothesis.
Although the results are very interesting, I am concerned about the sample size. If a sample size study was done, I think it would be appropriate to state this in the paper.
The high completion rate is 20.4%, leaving 118 professionals. I would like to know if these professionals are representative and/or if so, the statistical power of this sample.
Although at the end of the text something is said about the statistical analysis, in the Methodology section nothing is said about it. It is not clear what kind of statistical analysis is done.
I suppose that the Chi-Square test is used to find statistically significant differences in the sociodemographic variables between the total of those who start the course and those who finish the course.
Those who finish the course are in the group that started the course, so I consider it interesting to also compare these two groups: those who started but did not finish the course (460 professionals) with those who finished the course (118 professionals).
For this purpose, Table 1 should also include a description of the 460 professionals who started but did not finish the course.
In the discussion it is said that the high completion rate is large compared to other high completion rates. However, I think it is too small due to the fact that it is a free course.
80% of those who have completed the course are satisfied, but what do the professionals who started the course, but did not finish, think? Although this is a limitation of the study, I recommend that in the near future the opinion of the professionals who did not finish the course be taken into account.
Minor issues
The number of decimals in all tables and in the text should be unified. Put 1 or 2 decimals in the percentages and especially put 3 decimals in the p-values.
Author Response

(The authors gave the same response as above.)

Round 2
Reviewer 2 Report
Thank you very much for clarifying all my doubts.
There are no further comments.